# Enhanced Innate Immunity Mediated by IL-36α in Atopic Dermatitis and Differences in Cytokine Profiles of Lymphocytes in the Skin and Draining Lymph Nodes

**DOI:** 10.3390/biom15060817

**Published:** 2025-06-04

**Authors:** Ayaka Ichikawa, Mai Nishimura, Masako Ichishi, Yasutomo Imai, Yoshiaki Matsushima, Yoichiro Iwakura, Masatoshi Watanabe, Kiyofumi Yamanishi, Keiichi Yamanaka

**Affiliations:** 1Department of Dermatology, Mie University Graduate School of Medicine, 2-174 Edobashi, Tsu 514-8507, Mie, Japan; ichika-a@med.mie-u.ac.jp (A.I.); ishika-m@clin.medic.mie-u.ac.jp (M.N.); imaiyasutomo@gmail.com (Y.I.); matsushima-y@clin.medic.mie-u.ac.jp (Y.M.); kyamanis@me.com (K.Y.); 2Inflammatory Skin Disease Research Center, Mie University Graduate School of Medicine, 2-174 Edobashi, Tsu 514-8507, Mie, Japan; 3Department of Oncologic Pathology, Mie University Graduate School of Medicine, 2-174 Edobashi, Tsu 514-8507, Mie, Japan; masako-i@doc.medic.mie-u.ac.jp (M.I.); mawata@doc.medic.mie-u.ac.jp (M.W.); 4Imai Adult and Pediatric Dermatology Clinic, 5-1-1 Ebie, Fukushima, Osaka 553-0001, Osaka, Japan; 5Department of Biomedical Science, Graduate School of Agricultural and Life Science, The University of Tokyo, Tokyo 113-8657, Japan; iwakura@ims.u-tokyo.ac.jp

**Keywords:** alarmin, atopic dermatitis, interleukin-36α, inflammation, psoriasis

## Abstract

(1) Background: The IL-36 cytokines have been identified as key contributors to pustular psoriasis, and their inhibitor is already in clinical use. However, few studies have explored them in atopic dermatitis. (2) Methods: The role of IL-36α was investigated in various atopic dermatitis models using wild-type, keratin 14-specific IL-33 transgenic, IL-18 transgenic, caspase-1 transgenic, and caspase-1 transgenic mice with IL-17AF deletion, reflecting diverse aspects of human skin inflammation. IL-36α was administered subcutaneously in five doses on alternate days across the five strains to examine cellular infiltration patterns and cytokine expression levels. (3) Results: The skin phenotype was exacerbated, accompanied by worsening edema and skin thickness in all mouse groups upon IL-36α administration. An increase in infiltrating cells was observed among innate immune cells, while lymphocyte counts, including T cells and innate lymphoid cells, did not rise. Additionally, anti-inflammatory cytokines were induced simultaneously with inflammatory cytokines and downstream cytokines of IL-36α as well. Infiltrating lymphocytes in the skin displayed a distinct Type 2 cytokine-dominant profile for innate lymphoid cells and a Type 3 cytokine-dominant profile for T helper cells and γδ T cells, contrasting with the Type 1-dominant cell profile in draining lymph nodes. Type 1, Type 2, and Type 3 cytokine dominance patterns were not affected by the administration of IL-36α. (4) Conclusions: IL-36α triggers inflammatory responses in atopic dermatitis by activating innate immunity. The infiltrating lymphocytes in the skin have different cytokine production profiles between innate lymphoid cells and T cells, as well as different patterns of cytokine production in their draining lymph nodes.

## 1. Introduction

Inflammatory skin diseases are characterized by chronic inflammation, altered keratinocyte activity, and dysregulated immune function. In patients with psoriasis vulgaris and atopic dermatitis (AD), following the skin inflammation, an increased risk of cerebrovascular disease, ischemic heart disease, and arteriosclerosis, as well as a statistically shortened lifespan, has been reported [1,2]. Various studies have been conducted to support these findings, demonstrating that the overproduction of skin-derived inflammatory cytokines may lead to a range of organ failures, including cardio- and cerebrovascular disorders [3,4,5,6], systemic amyloidosis [5,7,8,9], impaired sperm motility [10], osteoporosis [11], diminished immune response in septic conditions [12], and even adipose tissue inflammation [13]. Therefore, skin inflammation needs to be controlled.

The interleukin (IL)-36 cytokines, members of the IL-1 superfamily, have emerged as critical players in the pathology of inflammatory skin diseases. Comprising three agonists (IL-36α, IL-36β, and IL-36γ) and two antagonists (IL-36Ra and IL-38), these cytokines interact with the IL-36 receptor (IL-36R) to activate inflammatory pathways that are essential for the initiation and maintenance of skin inflammation [14,15]. IL-36 aggravates neutrophil infiltration into the skin and pustular formation, and its inhibitors have already been employed in clinical situations [16,17,18,19,20]. Recent studies highlight IL-36’s role in enhancing pro-inflammatory cytokine secretion, promoting keratinocyte proliferation, and driving the recruitment and activation of diverse immune cells in the skin, thereby worsening conditions such as psoriasis [21,22]. Significantly higher serum IL-36α levels were reported in psoriasis arthritis patients [23]. However, the specific contributions of IL-36α, one of the most potent members of this cytokine family, in AD remain incompletely understood.

This research aims to delineate the effects of IL-36α on AD, by subjecting IL-36α to wild-type and genetically modified mice, which represent several patterns observed in human AD patients, we intend to comprehensively evaluate its role in different pathological conditions. Keratin 14-specific IL-33 overexpressing transgenic (IL-33Tg) mice: the mice develop skin-specific IL-33-induced acute-phase dermatitis with symptoms similar to human AD, keratin 14-specific IL-18 overexpressing transgenic (IL-18Tg) mice: models of chronic-phase AD, keratin 14-specific caspase-1 overexpression transgenic (KCASP1Tg) mice, which are regarded as a model that combines features of both AD and psoriasis [24]. Additionally, a mouse model excluding the IL-17A and F-mediated psoriatic response from KCASP1Tg mice (KCASP1Tg+IL-17AFKO) was utilized [25,26,27,28]. This study aims to confirm the role of IL-36α as an upstream regulator of the inflammatory cytokine cascade, as an alarmin, demonstrating its impact on various inflammatory pathways and establishing its significance in AD.

## 2. Materials and Methods

### 2.1. Mouse Models and Treatment

The current experiment included 50 female mice (4 or 6 months old), categorized into five groups: C57BL/6 (wild type: WT), IL-33Tg, IL-18Tg, KCASP1Tg, and KCASP1Tg+IL-17AFKO mice. The IL-33Tg mouse model, in which overexpression of IL-33 specific to epidermal keratinocytes leads to skin inflammation with features that resemble acute-phase AD, is based on its immunological profile and characteristic skin symptoms [28,29]. In IL-18Tg mice, skin symptoms develop at four months of age and peak at six months, and the chronic nature of inflammation is suggested by the observed immunological profile; this mouse has been proposed as a potential model for chronic-phase AD [30]. For IL-18Tg mice only, sampling was set at 6 months of age from the onset of the skin rash. KCASP1Tg mice develop a skin rash around the eyes starting at 8 weeks of age, which spreads to the entire face and trunk. Dermatitis presents symptoms of acute AD, including marked lymphocytic and eosinophilic infiltration, marked scratching behavior, and increased IgE [31]. However, there is also marked destruction of keratinocytes, and the cytokine pattern involved a Type 3 psoriasis profile in addition to Type 2 cytokines [24]. The KCASP1Tg+IL-17AFKO mouse model, which lacks IL-17A and F-related psoriasis elements, shows only AD elements. In each group, five mice were injected intradermally with IL-36α (4 μg per mouse each time; Biolegend, San Diego, CA, USA) dissolved in 0.5 mL of phosphate-buffered saline (PBS). The solution was injected five times every other day into the forehead area using a 30 G needle. Sample collection was performed two days after the final injection. As controls, five mice were injected with phosphate-buffered saline (PBS) on the same schedule. The area of skin erosion and ulcer was calculated using ImageJ JS version 1.54g (https://imagej.nih.gov/ij/index.html, accessed on 16 February 2025).

The Mie University Board Committee approved the experimental protocol for Animal Care and Use (#22-39-7-1, approval date: 20 May 2024). All mice were housed under specific environmental controls (temperature: 21 ± 2 °C, humidity: 60%, light cycle: 12/12 h) and allowed free access to food and water.

### 2.2. Tissue Sampling

All mice were subjected to euthanasia with CO_2_, and facial skin and cervical lymph nodes were collected. One-half of the skin was placed in a 10% formalin-neutral buffer solution (Wako, Osaka, Japan) and tissue freezing medium for histological and immunostaining analyses. The other half was used for flow cytometry analysis, or placed into tubes, snap-frozen, and then total RNA was extracted.

### 2.3. Histology and Immunohistochemistry

The skin samples embedded in paraffin were cut into 6 μm sections and stained with hematoxylin and eosin (H&E). In addition, immunohistochemical staining for CD4 (Cell Signaling Technology, Danvers, MA, USA), CD8 (Cell Signaling Technology), CD20 (Cell Signaling Technology), CD138 (Cell Signaling Technology), Ly6G (Abcam, Cambridge, UK), Iba-1 (Abcam, Cambridge, UK), and secondary antibody, goat anti-rabbit immunoglobulins/biotinylated (Dako, Santa Clara, CA, USA) and simple stain MAX-PO (rat) (Nichirei Biosciences, Tokyo, Japan) were performed to identify the specific type of infiltrating cells in the skin. Toluidine blue staining was also performed. Positive cells were counted in five random fields of view at ×400 magnification for each sample.

### 2.4. Flow Cytometry Analysis

Two square centimeters of skin per mouse were taken from five mice in each group. Cervical lymph nodes were also collected from five mice and were crushed in a petri dish, and the cells were filtered through a mesh and incubated with ACK Lysing Buffer (Thermo Fisher Scientific, Waltham, MA, USA) to lyse the red blood cells. The dermis was removed from the skin sample as much as possible, and the skin was cut into small pieces. The skin is crushed thoroughly for 5 s using an electric homogenizer. Then, the skin is immersed in RPMI 1640 containing 1% fetal calf serum, 85 mg/mL LiberaseTM (Roche, Basel, Switzerland), and 0.01% DNase I (Roche), and stirred vigorously for 60 min at 37 °C. Then, the mixture was passed through a 100 μm and 30 μm mesh. Further centrifuge for 1200 × 7 min, then add 50 mL of ACK for 15 min, centrifuge for 1200 × 7 min, and centrifuge again with 50 mL of PBS for 1200 × 7 min. Debris Removal Solution (Miltenyi Biotec, Auburn, CA, USA) was added and mixed well by pipetting 10 times. The cold PBS was supplemented gently over the layer, and centrifuged (4 °C, 4400 rpm, 10 min), leaving only the lower layer. The pellet was broken up by pipetting and dissolved in a cold buffer. Stir wash, then centrifuge at 4 °C at 2500 rpm for 10 min. Mononuclear cells were isolated and purified by density gradient centrifugation. These cells were incubated in the presence of phorbol myristate acetate (25 ng/mL), ionomycin (1 μg/mL), and brefeldin A (1 g/mL) for 4 h at 37 °C in an atmosphere of 5% CO_2_. The LIVE/DEAD^TM^ Fixable Aqua Dead Cell Stain Kit (Thermo Fisher Scientific) was used to exclude apoptotic and necrotic cells. The cultured mononuclear cells were stained with the following surface antibodies: CD45-APC, CD3-PE-Cy7, CD4- PE-Cy7, CD8a- PE-Cy7, CD45R-PerCp-Cy5.5, CD3-PerCp-Cy5.5, CD4-PerCp-Cy5.5, FceRI- PerCp-Cy5.5, CD8a-PerCp-Cy5.5, Ly-6G and Ly-6C-PerCp-Cy5.5, Siglec-F-PerCp-Cy5.5, TCRγδT--PerCp-Cy5.5, and BV786 hamster anti-mouse γδT cell receptor (BD Biosciences, Franklin Lakes, NJ, USA) in cell surface staining buffer containing 0.1 M PBS and 1% bovine serum albumin (BSA; Sigma-Aldrich, St. Louis, MO, USA), and then stained with IL-13-fluorescein isothiocyanate (FITC), IL-17A-APC-Cy7, IL-17F-phycoerythrin (PE), IFNγ-Brilliant Violet 605, and Brilliant Violet 421 IL-5 antibodies (BD Biosciences). The expression patterns of inflammatory cytokines were analyzed using a BD Lyric flow cytometer (BD Biosciences), and data were analyzed using FlowJo software (v10.10.0, Tree Star Inc., Ashland, OR, USA).

### 2.5. Real-Time Polymerase Chain Reaction (Real-Time PCR)

RT-PCR was performed to measure the changes in mRNA levels in the facial skin at the injection site. Total RNA was extracted using Tri Reagent (Molecular Research Center, Cincinnati, OH, USA). The RNA concentration was measured using a NanoDrop Lite spectrophotometer (Thermo Fisher Scientific, Waltham, MA, USA). Approximately 1 μg of total RNA was converted to cDNA using a High-Capacity RNA-to-cDNA Kit (Applied Biosystems, Waltham, MA, USA). The TaqMan Universal PCR Master Mix II with UNG (Applied Biosystems) was used to measure the mRNA expression of TNFα (Mm00443258_m1), IFNγ (Mm01168134_m1), IL-17A (Mm00439618_m1), IL-17F (Mm00521423_m1), IL-4 (Mm00445259_m1), IL-5 (Mm00439646), IL-13 (Mm00434204_m1), TGFβ (Mm03024053_m1), IL-10 (Mm99999062_m1), and IL-27 (Mm00461164_m1). The downstream cytokines of IL-36α were also measured using specific primers as follows: IL-6 (Mm00446190_m1), Intercellular Adhesion Molecule 1 (Icam1, Mm00516023_m1), C-C Motif Chemokine Ligand 2 (Ccl-2, Mm00441242_m1), Early Growth Response 1 (Egr1, Mm00656724_m1), c-Fos (Fos, Mm00487425_m1), Matrix Metalloproteinase-9 (Mmp-9, Mm00442991_m1), Cyclooxygenase-2 (COX2, Mm03294838_g1), loricrin (Mm01962650_s1), involucrin (Mm00515219_s1), and filaggrin (Mm01716522_m1). Glyceraldehyde-3-phosphate dehydrogenase (GAPDH, Mm99999915_g1) was used as an internal control. All probes were purchased from Thermo Fisher Scientific, and the amplification was performed in a LightCycler 96 System (Roche Diagnostics, Indianapolis, IN, USA). The cycling parameters were as follows: 50 °C for 120 s, 95 °C for 600 s, followed by 40 cycles of amplification at 95 °C for 15 s, and 60 °C for 60 s.

### 2.6. Statistical Analysis

Statistical analyses were performed using Prism software version 10 (GraphPad, San Diego, CA, USA). All groups were analyzed using the Mann–Whitney U test. Differences were considered statistically significant at * *p* < 0.05 and ** *p* < 0.01.

## 3. Results

### 3.1. Administration of IL-36α Causes Exacerbation of Cutaneous Manifestations

WT did not develop a skin symptom but developed skin inflammation after IL36α administration, similar to the previous report [32]; dermatitis showed erosion formation, crusting, and mild lichenification around the eyes and nose (Figure 1A). In IL-33Tg, erythema with exudation, crusts, scales, and hair loss were observed around the eyes, periorbital and perinasal regions of the face, and ears, even without IL36α injection. After IL36α administration, IL-33Tg mice showed increased skin thickening, edema, and erosion. IL-18Tg is a mouse model of AD of the chronic phase, and erosive dermatitis, reepithelization, and lichenoid changes were observed. IL36α administration resulted in marked lichenification with crusting. Thickening and loss of skin elasticity made opening their eyes difficult for some of them. In KCASP1Tg, even without IL36α injection, erosion started from the face and extended to the ear and neck. Multiple ulcers were formed on the face. Reepithelialization with atrophic skin occurred, but erosion and ulceration quickly relapsed. The hair of the face and eyelids decreased, and the eyes were covered with a fibrous membrane. In KCASP1Tg with IL36α injection, erosions enlarged over the entire face, with marked scars and skin thickening. Mice lacking IL-17AF from KCASP1Tg exhibited a milder overall lower degree of inflammation than KCASP1Tg; after IL36α administration, KCASP1Tg+IL-17AFKO showed crust formation mainly around the eyes and on the face. The area of skin lesions increased in all groups when IL-36α was administered compared to the non-treated group. For reference, the area of skin lesions was predominantly larger in the KCASP1Tg group compared to other groups (Figure 1B).

### 3.2. IL-36α Induces the Recruitment of Innate Immune Cells to the Skin Tissue

The IL-36α non-treated whole mouse models, except WT, displayed inflammatory responses characterized by mild epidermal thickening and immune cell infiltration into the dermis, as observed through H&E staining (Figure 2A). The number of CD4 and CD8 T cells in these mice tends to increase, especially in IL-18Tg, even in the IL-36α non-treated mice. However, infiltration of Ly6G-positive neutrophils and Iba-1-positive monocytes/macrophages was not prominent. KCASP1Tg and KCASP1Tg+IL-17AFKO exhibited an increase in CD138-positive cells (Figure 2B,C).

After IL-36α administration, distinct changes were noted in each mouse model. In WT, HE staining showed a marked increase in immune cell infiltration in the deeper layers of the dermis. CD4-positive T cells, CD8-positive T cells, and CD20-positive B cells increased significantly. CD138 staining revealed increased plasma cells, while Ly6G staining confirmed an increase in neutrophils. Iba1-positive monocytes/ macrophages were also increased. In IL-33Tg, immune cell infiltration was primarily observed in the superficial layers of the dermis, as demonstrated by H&E staining. Iba1 staining showed significant increases in monocytes and macrophages. CD138 and Ly6G also showed increases with statistical significance. Toluidine blue staining indicated an increase in mast cells, although this increase was not statistically significant. For IL-18Tg, immune cell infiltration was predominantly found in the deeper layers of the dermis. CD138, Ly6G, and Iba1 staining positive cells revealed increases, indicating heightened infiltration of plasma cells, neutrophils, and macrophages, respectively. In KCASP1Tg, widespread multiple types of immune cell infiltration were observed throughout all layers of the dermis. Finally, in KCASP1Tg+IL-17AFKO, immune cell infiltration increased across all dermal layers, as shown by H&E staining. Neutrophils increased, with Ly6G staining revealing a noticeable increase (Figure 2C,D).

### 3.3. Pro-Inflammatory and Anti-Inflammatory Cytokines Are Increased by IL-36α in WT and IL-18Tg Mice

After IL-36α administration, RT-PCR analysis revealed that IFNγ expression was increased in WT. Type 2 cytokines such as IL-4, IL-5, and IL-13 tended to increase. Although not statistically significant, RT-PCR analysis showed a decrease in TNFα expression following IL-36α administration in all mouse models. IFNγ expression was increased in IL-18Tg, while IL-4, IL-5, IL-13, and IL-17F levels also showed an increasing trend in IL-18Tg after IL-36α injection. KCASP1Tg and KCASP1Tg+IL-17AFKO showed no predominant changes (Figure 3A). After binding of IL-36α to its receptor, through NF-κB or MAPK pathway activation, the downstream cytokines, chemokines, and transcription factors may be enhanced. In the context of NF-κB signaling, IL-6 was increased in IL-18Tg mice, while CCL2 was elevated in both WT and IL-33Tg mice. Regarding MAPK signaling, EGR1 was decreased in WT mice, and c-Fos was reduced in WT, IL-18Tg, and KCASP1Tg+IL-17AFKO mice. Furthermore, MMP-9 was increased in IL-18Tg mice, whereas COX-2 was decreased in WT and KCASP1Tg±IL-17AFKO mice (Figure 3B).

Based on these findings, additional investigations were conducted on the anti-inflammatory cytokines TGFβ, IL-10, and IL-27 (Figure 4). These cytokines were examined to understand their potential role in suppressing cellular activity. TGFβ was induced in IL-33Tg after IL-36α injection. The anti-inflammatory cytokines IL-10 and IL-27 showed an increasing trend after IL-36α administration in all strains. IL-27 exhibited notable changes in WT and IL-18Tg following IL-36α injection.

To better understand and visualize the overall immunological differences and IL-36α responsiveness among the models, a summary table (Table 1) was provided, integrating their inflammatory profiles, disease relevance, and post-treatment immune changes.

### 3.4. IL-36α Enhances Its Own Expression and Is Associated With Barrier Gene Downregulation

To investigate how IL-36α influences cytokine expression and epidermal differentiation, we performed RT-PCR analysis of IL-36 isoforms (Il36a, Il36b, Il36g) and barrier-related genes (loricrin, involucrin, filaggrin) in five mouse strains with and without IL-36α treatment. As shown in Figure 5A, Il36a and Il36b expression was significantly increased in IL-18Tg mice following IL-36α administration, while other strains showed a similar but non-significant trend. In contrast, Il36g expression was significantly decreased in WT mice and tended to decrease in the other strains as well. In addition to IL-36 isoforms, we quantified mRNA levels of barrier-related genes to assess epidermal differentiation. Among them, loricrin was consistently downregulated across all strains, while involucrin and filaggrin showed no statistically significant changes, although mild strain-dependent variations were observed (Figure 5B).

### 3.5. Trend of Th17 and ILC3 Predominance in the Skin and Th1 Predominance in the Lymph Node

The cytokine production profiles of lymphocytes in the skin and the lymph nodes of WT and AD model mice were examined using flow cytometric analysis. In the skin, WT, IL-33Tg, IL-18Tg, and KCASP1Tg mice predominantly expressed IL-17-producing cells in CD3+4+8a+ T cells and γδ T cells, with KCASP1Tg having a higher number of Th17 cells (Figure 6A,B). In contrast, the percentage of Type 2 cytokine-producing cells was dominant in the ILC population in all strains. After IL-36α administration, a trend toward a decrease in the total number of CD3+4+8a+ T cells in the skin of IL-33Tg, IL-18Tg, and KCASP1Tg was observed (Figure 6A,B). Similarly, γδ T cell counts in the skin decreased across all models. The ILC counts in the skin also tended to decrease in IL-33Tg, KCASP1Tg, and KCASP1Tg+IL-17AFKO. This cytokine profile was not altered by IL36α administration. In KCASP1Tg+IL-17AFKO, Type 2 cells were predominant compared to Type 1 cells.

Conversely, in the lymph nodes, all mouse models showed a trend toward a higher percentage of Type 1 cells in CD3+4+8a+ T cells, γδ T cells, and ILCs compared to the skin. In all strains of mice, administration of IL-36α did not significantly alter the cytokine balance.

## 4. Discussion

IL-36 is a member of the IL-1 superfamily and is widely expressed in cells of the skin, lungs, and intestines, where it plays a critical role in innate immunity and inflammatory responses [33,34,35]. IL-36 receptor agonists (IL-36α, IL-36β, IL-36γ) are rapidly secreted by keratinocytes, epithelial cells, and immune cells in response to inflammatory stimuli like lipopolysaccharides (LPS) and adenosine triphosphate (ATP) [36,37]. Once activated, IL-36 binds to IL-36R and triggers downstream signaling pathways, such as MAPK and NF-κB, leading to the expression of pro-inflammatory mediators, including TNF-α, IL-1β, IL-6, and antimicrobial proteins like S100A7 and defensins [38,39,40,41]. For example, during *Staphylococcus aureus* exposure, IL-36α secretion from keratinocytes induces skin inflammation characterized by the infiltration of γδT cells, CD4+ T cells, and neutrophils via MyD88 signaling [42,43]. Additionally, IL-36 influences Th1/ Th17 polarization and enhances cytokine cascades [44,45,46]. These mechanisms underscore its pivotal role in inflammatory skin diseases as a driver of early immune activation and disease progression.

The initial inflammatory response triggered by IL-36 also includes the induction of anti-inflammatory molecules such as IL-36Ra and IL-38, which function as feedback mechanisms to regulate inflammation [47]. In pustular psoriasis, mutations in the IL36RN gene, encoding IL-36Ra, result in its dysfunction, exacerbating downstream inflammatory cascades [48]. Beyond IL-36Ra and IL-38, cytokines like TGFβ, IL-10, and IL-27 may also be upregulated in response to IL-36 induction [49,50]. TGFβ promotes regulatory T cell differentiation, leading to IL-10 production and immune suppression [51]. Similarly, IL-27 induces IL-10 expression in Th17 cells through the STAT3 pathway and plays dual roles as both a pro-inflammatory and anti-inflammatory cytokine [52,53]. Together, these molecules contribute to the regulation of inflammation and highlight the complex feedback mechanisms modulating IL-36-driven immune responses.

This study evaluated the effects of IL-36α overload in WT mice, as well as in multiple AD models exhibiting diverse inflammatory characteristics, a model combining features of AD and psoriasis, and an AD model with a deleted Type 3 cytokine response. Following IL-36α administration, skin phenotype changes were accelerated in all strains (Figure 1), with increased infiltration of plasma cells, neutrophils, and macrophages into the skin tissues of all models (Figure 2). This increase represented a marked amplification of cells associated with innate immunity. Conversely, an increase in infiltrating cells from the adaptive immune system, such as CD4+ cells, was observed only in WT mice, not in other inflammatory models where inflammation was already present. This suggests that IL-36α acts as an initiator of inflammation across various models.

Cytokine analysis highlighted IL-36α’s ability to enhance pro-inflammatory signals, which subsequently triggered regulatory feedback mechanisms. RT-PCR results revealed an increase in IFNγ in WT and IL-18Tg mice following IL-36α injection, while TNFα levels showed a declining trend in WT. Among Type 1 cytokines, a balance exists between TNFα and IFNγ, with TNFα potentially decreasing when IFNγ is predominant, as observed in this study (Figure 3A) [54]. Analyses of MAPK and NF-κB pathways revealed a complex modulation of downstream targets following IL-36α administration. IL-36α predominantly activated the NF-κB signaling pathway, as evidenced by an increase in IL-6 in IL-18Tg mice and elevated CCL2 levels in WT and IL-33Tg mice. In contrast, IL-36α appeared to suppress the MAPK pathway, as indicated by reduced EGR1 expression in WT mice, decreased c-Fos levels in WT, IL-18Tg, and KCASP1Tg+IL-17AFKO mice, and downregulated COX-2 in WT and KCASP1Tg+IL-17AFKO mice (Figure 3B). IL-36α injection also activated anti-inflammatory cytokine production as a feedback mechanism (Figure 4). The notable increase in IL-27 observed in IL-18Tg mice following IL-36α administration aligns with the pronounced inflammatory cell infiltration in these mice. This finding reinforces the idea that IL-27 and IL-10 production act as compensatory feedback mechanisms, balancing the heightened pro-inflammatory activity driven by IL-36α. To improve translational relevance, we compared the immunological characteristics summarized in Table 1 with known findings in human AD. In patients with AD, lesional skin typically exhibits Th2- and Th1-dominated inflammation, with elevated levels of IL-4, IL-13, and IFN-γ, as well as increased infiltration of CD4⁺ T cells, eosinophils, and macrophages [55,56]. IL-18Tg mice reflect these immune profiles, with post-IL-36α administration resulting in marked infiltration of Ly6G⁺ neutrophils, Iba1⁺ macrophages, and CD138⁺ plasma cells (Figure 2), along with increased expression of key cytokines such as IFN-γ, IL-13, and IL-4 (Figure 3). Additionally, WT mice showed induction of CCL2 and IL-10 after IL-36α, mirroring chemokine and regulatory cytokine responses reported in human AD lesions. These parallels suggest that the murine models used in the current study accurately reproduce key immune cell and cytokine features observed in human AD skin, reinforcing the relevance of IL-36α-driven responses in the disease pathogenesis. In addition to immune modulation, IL-36α also affected epidermal gene expression (Figure 5). IL-36α and IL-36β were significantly upregulated in IL-18Tg mice following exogenous IL-36α administration, suggesting that IL-36α can induce its own gene expression. This is consistent with increased IL-36α levels observed in lesional skin of AD patients [57]. Among barrier-related genes, loricrin was consistently downregulated across all strains, indicating impaired terminal keratinocyte differentiation. Involucrin and filaggrin showed no significant changes. Similar alterations in barrier-related molecules and biomarker expression have been reported in both AD and psoriasis [58]. Additionally, in inflammation-prone models, such as KCASP1Tg and IL-18Tg, IL-36α administration led to a reduction in absolute lymphocyte numbers in the lesional skin (Figure 6A). This reduction may be attributable to an increase in anti-inflammatory cytokines, highlighting the interplay between pro-inflammatory and regulatory responses. Together, these results suggest that IL-36α modulates immune cell dynamics through both direct and feedback mechanisms, actively shaping the immune environment in these models.

What was surprising in the analysis of lymphocytes infiltrating the skin was that ILCs produced more Type 2 cytokines, while T helper cells produced more Type 3 cytokines. Different roles were found among the cells infiltrating the lesional skin in the same strain of mice. Another novel finding was that the inflammatory cells in the skin were predominantly Type 2 or Type 3 cytokine-producing cells; conversely, in the lymph nodes, CD3+4+8a+ T cells primarily exhibited a Type 1 profile, with γδ T cells and ILCs showing higher proportions of Type 1 and Type 3 cells. These findings suggest distinct immune responses occurring in the skin compared to the lymph nodes.

The KCASP1Tg+IL-17AFKO model provides further insights into modulating immune profiles. KCASP1Tg+IL-17AFKO mice demonstrated an increased proportion of Type 2 cells compared to WT mice in the lesional skin, aligning with clinical observations that IL-17 blockade in psoriasis patients shifts the cytokine environment from a psoriasis-like inflammatory profile to one more characteristic of AD [59,60]. Although the experiment was conducted in mice, examining the cytokine balance of lymphocytes in vivo shows that the immune environment is generally dominated by Type 1 cytokine-producing cells, with a subtle balance between Types 2 and 3 [61]. Consequently, more aggressive IL-17 blockade in psoriasis treatment is likely to shift the immune balance toward Type 2 dominance [61]. In KCASP1Tg+IL-17AFKO(+), Type 2 cells increased markedly compared to KCASP1Tg(+), mirroring observations in clinical settings.

The term “alarmin” was introduced in the early 2000s to describe molecules released by damaged cells that act as danger signals to activate the immune system [62]. Initially, alarmins were defined as endogenous molecules rapidly released during necrosis or cellular stress to initiate immune responses. Early examples included molecules like High Mobility Group Box 1 (HMGB1) and cathelicidin, which are released in response to injury and activate immune cells. Over time, the definition expanded to encompass proteins secreted in response to inflammatory stimuli, such as IL-33 and S100 proteins [63,64]. Alarmins are now recognized as factors released during tissue damage that function as endogenous molecules promoting immune activation, playing roles in infection defense, tissue repair, and immune balance. IL-36 has been shown to be rapidly released in response to damage or infection, demonstrating potent properties in amplifying both innate and adaptive immune responses [65]. In this context, IL-36, which is released from keratinocytes during cell damage and acts on the immune system, can be considered an alarmin.

## 5. Conclusions

In conclusion, our studies demonstrate that IL-36α primarily mobilizes innate immune cells across a variety of inflammatory pathways, without a predominant increase in acquired lymphocyte production. This mobilization was accompanied by an increased feedback mechanism involving anti-inflammatory cytokines. Among skin-infiltrating lymphocytes, T helper cells and γδT cells primarily produced Type 3 cytokines, whereas ILCs predominantly produced Type 2 cytokines. In contrast, lymph nodes had more Type 1 cytokine-producing cells, and cytokine profiles differed markedly between the skin and their lymph nodes; administration of IL-36α did not alter these established balances.

## Figures and Tables

**Figure 1 biomolecules-15-00817-f001:**
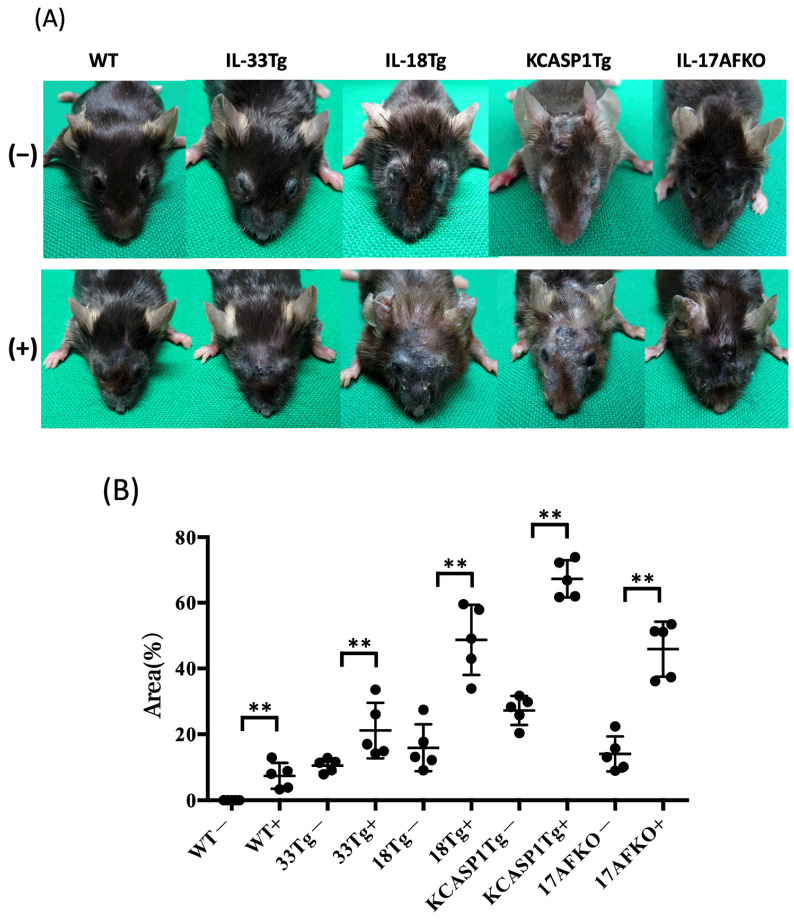
(**A**) Clinical features of WT, IL-33Tg, IL-18Tg, KCASP1Tg, and KCASP1Tg+IL-17AFKO mice, with or without IL-36α treatment. “(–)” indicates the IL-36α-untreated group; “(+)” indicates the IL-36α-treated group. WT mice did not develop skin eruptions under baseline conditions; however, IL-36α injection induced cutaneous inflammation, characterized by mild erosions and lichenification, particularly around the eyes and nose. In IL-33Tg mice, erythema with exudation, erosions, crusts, scaling, and alopecia were observed around the periorbital and perinasal regions, as well as the ears, even without IL-36α treatment. Upon IL-36α administration, IL-33Tg mice exhibited exacerbated skin thickening, erosions, and edema. In IL-18Tg mice, a chronic AD model, severe erosive dermatitis, reepithelialization, and lichenoid inflammation were evident. IL-36α treatment further induced pronounced lichenification and crusting. Skin thickening and loss of elasticity rendered eye opening difficult in some mice. In KCASP1Tg mice without IL-36α, erosions originated on the face and spread to the ears and neck. Multiple ulcers developed on the face, trunk, and extremities. Although reepithelialization with atrophic skin occurred, erosions and ulcers frequently relapsed. Facial and eyelid hair was lost, and a fibrotic membrane covered the eyes. In IL-36α-treated KCASP1Tg mice, erosions enlarged to cover the entire face, accompanied by pronounced scarring and skin thickening. KCASP1Tg mice lacking IL-17AF (KCASP1Tg+IL-17AFKO) developed milder inflammation than KCASP1Tg mice. Upon IL-36α treatment, these mice exhibited localized erosions and crusts, predominantly around the eyes and face. Representative images of mice from each group are shown. (**B**) The area of facial and neck skin erosion and the ulcer was calculated using ImageJ JS. The percentage of area of skin lesions was increased in all groups when IL-36α was administered, compared to the non-treated group. The area of skin lesions was larger in the KCASP1Tg group than in the other groups (** *p* < 0.01).

**Figure 2 biomolecules-15-00817-f002:**
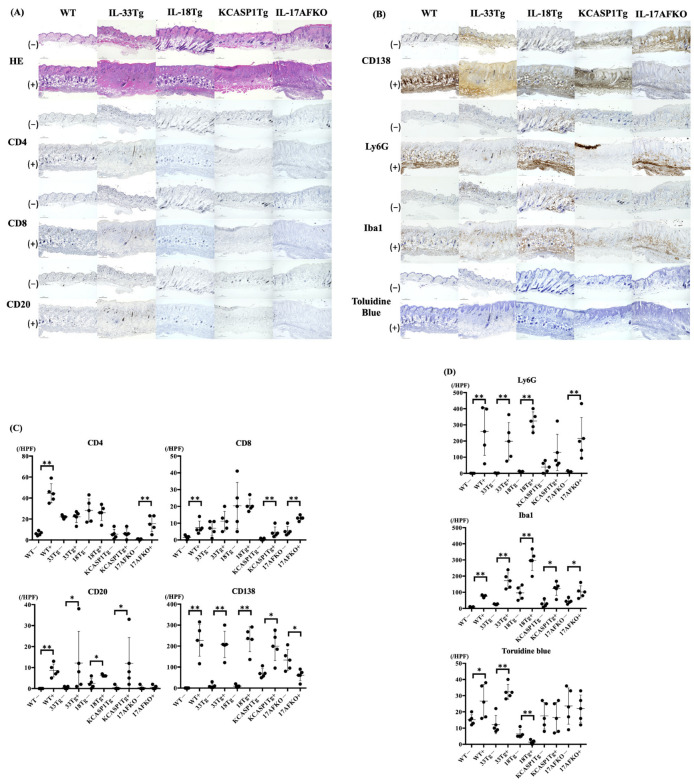
(**A**) H&E and CD4, CD8, CD20 immunohistochemical stainings of skin tissue from WT, IL-33Tg, IL-18Tg, KCASP1Tg, and KCASP1Tg+IL-17AFKO mice, observed at 10× magnification (scale bar 200 μm). All strains of mice, but not the WT group, showed large infiltration of immune cells even in IL-36α non-treated groups. After IL-36α administration, all mouse models exhibited massive immune cell infiltration and epidermal thickening. (**B**) CD138, Ly6G, and Iba1 immunohistochemical staining and toluidine blue staining of skin tissue from the same mouse models, observed at 10× magnification (scale bar 200 μm). CD138, Ly6G, and Iba1 staining showed the presence of plasma cells, neutrophils, and monocytes or macrophages, respectively. CD138 and Ly6G positive cells were increased significantly in IL-36α-administrated groups. (**C**,**D**) Quantitative analysis of immune cell infiltration. Positive cells were counted at five randomly selected fields (400× magnification) per sample. In the IL-36α administration group, WT mice showed an increase in all types of cells, while IL-33Tg mice exhibited an increase in CD20, CD138, Ly6G, Iba1, and toluidine blue-positive cells. IL-18Tg mice demonstrated increases in CD20, CD138, Ly6G, Iba1, and toluidine blue-positive cells. KCASP1Tg mice showed an increase in CD8, CD20, CD138, and Iba1-positive cells, and KCASP1Tg+IL-17AFKO mice had an increase in CD4, CD8, CD138, Ly6G, and Iba1-positive cells. All groups were analyzed using the Mann–Whitney U test (* *p* < 0.05; ** *p* < 0.01).

**Figure 3 biomolecules-15-00817-f003:**
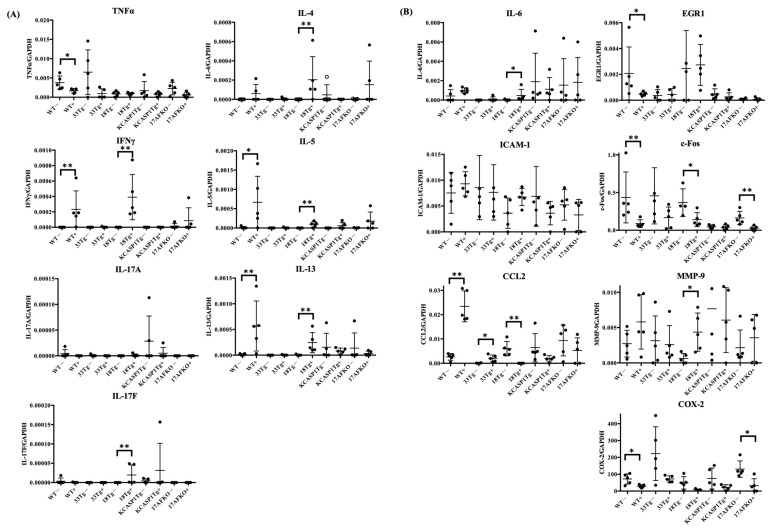
(**A**) Expression levels of pro-inflammatory cytokines (TNFα, IFNγ, IL-4, IL-13, IL-17A, and IL-17F) were assessed in skin samples from WT, IL-33 transgenic (IL-33Tg), IL-18 transgenic (IL-18Tg), KCASP1 transgenic (KCASP1Tg), and KCASP1Tg+IL-17AF knockout (KCASP1Tg+IL-17AFKO) mice, with or without IL-36α treatment. RT-PCR analysis revealed upregulation of Type 2 cytokines, including IL-5 and IL-13, following IL-36α administration in WT mice. In contrast, Type 1 cytokines exhibited differential responses: TNFα was constitutively expressed but tended to decrease, possibly due to a shift toward Type 2 cytokine induction; IFNγ expression showed a modest increase. IL-17AF levels remained unchanged in WT mice. In IL-18Tg mice, IL-36α administration resulted in increased expression of IFNγ, IL-4, IL-5, and IL-13. No notable changes were observed in KCASP1Tg or KCASP1Tg+IL-17AFKO mice. (**B**) Gene expression associated with NF-κB and MAPK signaling pathways following IL-36α treatment. Within the NF-κB pathway, IL-6 expression was significantly elevated in IL-18Tg mice, while CCL2 levels increased in both WT and IL-33Tg mice. Regarding the MAPK pathway, EGR1 expression was significantly downregulated in WT mice, and c-Fos expression was reduced in WT, IL-18Tg, and KCASP1Tg+IL-17AFKO mice. MMP-9 expression was markedly upregulated in IL-18Tg mice, whereas COX-2 expression was significantly suppressed in both WT and KCASP1Tg+IL-17AFKO mice. Statistical significance was assessed using the Mann–Whitney U test (* *p* < 0.05; ** *p* < 0.01).

**Figure 4 biomolecules-15-00817-f004:**
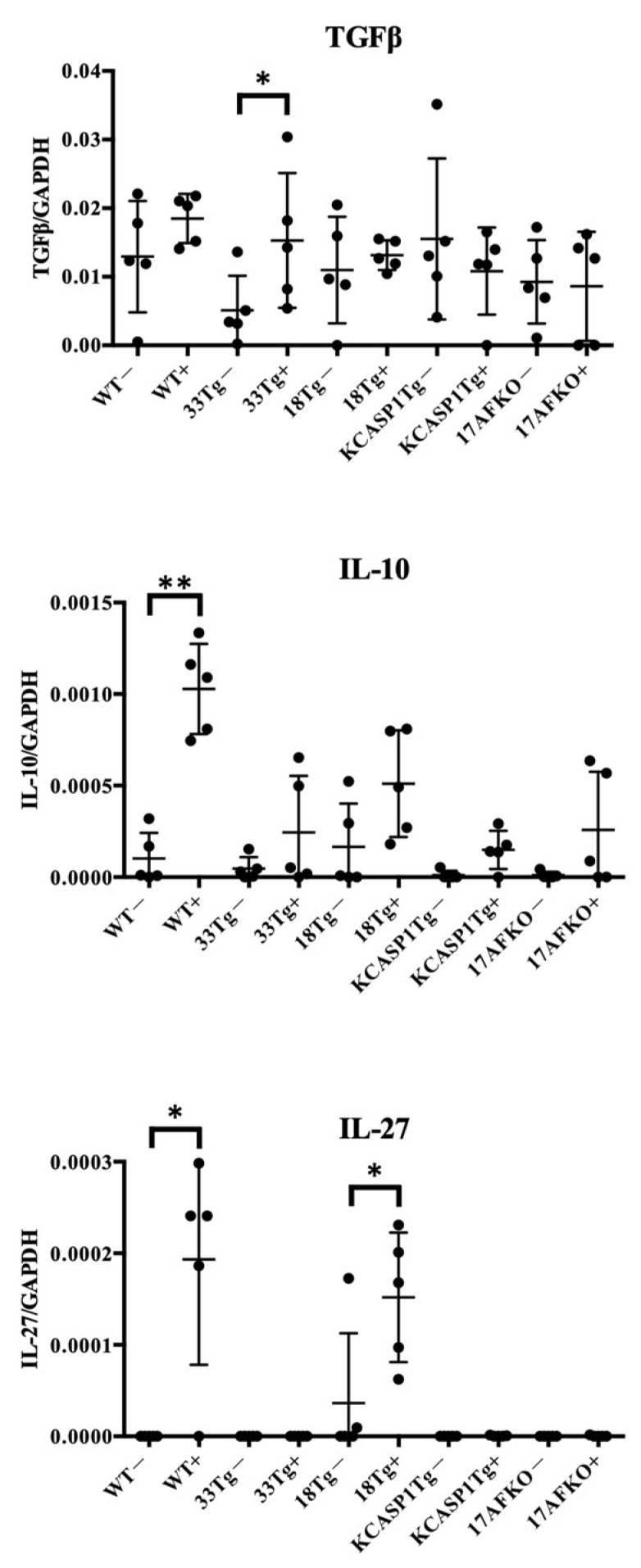
The expression levels of anti-inflammatory cytokines (TGFβ, IL-10, and IL-27) were measured in skin samples from the same groups of mice. TGFβ was increased in IL-33Tg after IL-36α injection. After IL-36α administration, IL-10 showed an increasing trend in whole strains, but no statistically significant increase was observed in any of the AD mouse models. IL-27 levels were increased in WT and IL-18Tg following IL-36α injection. Statistical significance was determined using the Mann–Whitney U test (* *p* < 0.05; ** *p* < 0.01).

**Figure 5 biomolecules-15-00817-f005:**
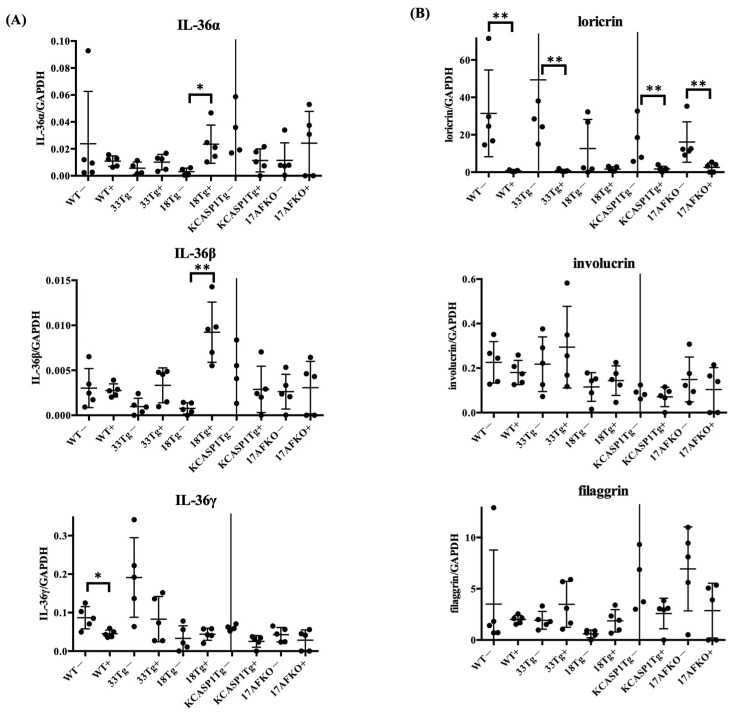
(**A**) Expression levels of IL-36 isoforms (Il36a, Il36b, and Il36g) were analyzed in skin samples from WT, IL-33 transgenic (IL-33Tg), IL-18 transgenic (IL-18Tg), KCASP1 transgenic (KCASP1Tg), and KCASP1Tg+IL-17AF knockout (KCASP1Tg+IL-17AFKO) mice, with or without IL-36α treatment. Following IL-36α administration, Il36a and Il36b expression was significantly upregulated in IL-18Tg mice. In contrast, Il36g expression was significantly downregulated in WT mice and tended to decrease in the other strains. (**B**) Expression levels of epidermal barrier-associated genes (loricrin, involucrin, and filaggrin) were evaluated in the same set of samples. Loricrin expression was consistently reduced across all strains after IL-36α treatment. However, no statistically significant changes were observed in the expression of involucrin or filaggrin in any strain. Statistical significance was determined using the Mann–Whitney U test (* *p* < 0.05; ** *p* < 0.01). In some groups, error bars extend beyond the y-axis limits due to extreme values and are not visible in the graph.

**Figure 6 biomolecules-15-00817-f006:**
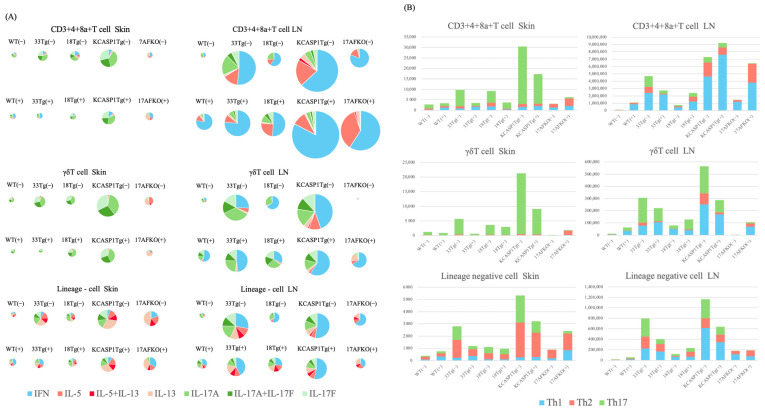
Quantification of cytokine-producing cells in the skin and lymph nodes of various mouse models. (**A**) Pie charts illustrating the proportions of Type 1, Type 2, and Type 3 cytokine-producing cells in skin and lymph node (LN) samples from WT, IL-33Tg, IL-18Tg, KCASP1Tg, and KCASP1Tg+IL-17AFKO mice, with and without IL-36α treatment. The analysis was conducted on CD3⁺CD4⁺CD8a⁺ T cells, γδ T cells, and innate lymphoid cells (ILCs). Colored segments represent cells producing IFNγ, IL-5, IL-5+IL-13, IL-13, IL-17A, IL-17A+IL-17F, and IL-17F. The overall size of each pie chart reflects the total number of cells identified. In the skin, CD3⁺CD4⁺CD8a⁺ T cells and γδ T cells in WT, IL-33Tg, IL-18Tg, and KCASP1Tg mice primarily expressed IL-17A and IL-17F, with KCASP1Tg mice exhibiting the highest abundance of Th17 and γδ T cells. Notably, Type 2 cytokine-producing ILCs were proportionally increased in these mice. In KCASP1Tg+IL-17AFKO mice, Type 2 cytokine-producing γδ T cells and ILCs were predominant over Type 1 cells. In contrast, in the LN, CD3⁺CD4⁺CD8a⁺ T cells in all models exhibited a relative increase in Type 1 cytokine production compared to the skin. Among γδ T cells, Type 3 cytokine production remained dominant, although IFNγ-producing cells were also elevated. Within the ILC population, Type 2 cytokine production decreased, while IFNγ-producing cells increased across models, compared to those in the skin profiles. (**B**) Bar graphs showing the absolute numbers of Type 1, Type 2, and Type 3 cytokine-producing cells in the skin and LN across the same mouse models before and after IL-36α treatment. Data represent mean values from pooled samples analyzed by flow cytometry. In the skin, IL-36α administration resulted in a reduction of CD3⁺CD4⁺CD8a⁺ T cells in IL-33Tg, IL-18Tg, and KCASP1Tg mice. γδ T cell numbers decreased across all models, and ILC numbers also tended to decline in IL-33Tg, KCASP1Tg, and KCASP1Tg+IL-17AFKO mice. Conversely, in the LN, an increase in CD3⁺CD4⁺CD8a⁺ T cells was observed in WT, IL-18Tg, KCASP1Tg, and KCASP1Tg+IL-17AFKO mice. γδ T cells were also increased in the LN of WT, IL-18Tg, and KCASP1Tg+IL-17AFKO mice following IL-36α treatment.

**Table 1 biomolecules-15-00817-t001:** Summary of mouse models and their immune responses to IL-36α. This table summarizes the immunological characteristics and IL-36α-induced responses observed in each mouse model. Inflammation type (Type 1, Type 2, Type 3) and disease relevance (AD-like or psoriasis-like) are indicated for each strain. Post-treatment changes in immune cell infiltration and cytokine expression are shown only for statistically significant results. Abbreviations: TB, toluidine blue; ↑, increase; ↓, decrease; n.s., not significant. Only changes with *p* < 0.01 are listed explicitly. Changes that showed significance at *p* < 0.05 but not at *p* < 0.01 are denoted as “–”. Non-significant changes (*p* ≥ 0.05) are indicated as “n.s.”.

Mouse Model	Inflammation Type	Disease Model	Post-IL-36 Immune Cells	Post-IL-36 Cytokines
WT	None	Control	↑ CD4⁺, ↑ CD8⁺, ↑ CD20⁺, ↑ CD138⁺, ↑ Ly6G⁺, ↑ Iba1⁺	↑ IFN-γ, ↑ IL-13, ↑ CCL2, ↓ c-Fos, ↑ IL-10
IL-33Tg	Type 2	Acute-phase AD	↑ CD138⁺, ↑ Ly6G⁺, ↑ Iba1⁺, ↑ TB⁺	–
IL-18Tg	Type 2	Chronic-phase AD	↑ CD138⁺, ↑ Ly6G⁺, ↑ Iba1⁺, ↓ TB⁺	↑ IFN-γ, ↑ IL-4, ↑ IL-5, ↑ IL-13, ↑ IL-17F, ↓ CCL2
KCASP1Tg	Type 2/Type 3	Mixed AD / Psoriasis-like	↑ CD8⁺	n.s.
KCASP1Tg + IL-17AFKO	Type 2	AD (Type 3 -independent)	↑ CD4⁺, ↑ CD8⁺, ↑ Ly6G⁺	↓ c-Fos, ↓ COX-2

## Data Availability

The raw data supporting the conclusions of this article will be made available on request.

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
