# Peer review of "Enhanced Innate Immunity Mediated by IL-36α in Atopic Dermatitis and Differences in Cytokine Profiles of Lymphocytes in the Skin and Draining Lymph Nodes"

_biomolecules, 2025, doi:10.3390/biom15060817_

Round 1
Reviewer 1 Report
Comments and Suggestions for Authors
Dear Authors,
The submitted manuscript has scientific soundness and significance; however, there are several flaws in the manuscript that need to be addressed. The manuscript cannot be considered for publication in the journal in its current form. My comments are attached herewith, which can be addressed pointwise, and revise the manuscript accordingly.

Author Response
Dear Authors,
The submitted manuscript has scientific soundness and significance; however, there are several flaws in the manuscript that need to be addressed. The manuscript cannot be considered for publication in the journal in its current form. My comments are attached herewith, which can be addressed pointwise, and revise the manuscript accordingly.
Comments
- This research work thoroughly investigates the immunological function of IL-36α in
atopic dermatitis (AD) by contrasting its effects in many genetically modified and wildtype mice models that mimic the acute and chronic stages of AD and situations similar to psoriasis. The overall study performed is very interesting, however manuscript needs to be improves as suggested.
Response: Thank you for taking the time to review our paper. We have prepared a revision reflecting the reviewers' remarks. Thank you for your cooperation.
- The authors assume that readers are familiar with the relevance of each mouse model to human AD or psoriasis. Please provide a brief comparison chart highlighting the immunopathological characteristics of each strain will increase understanding for a
wider audience.
Response: Thank you for your advice. We have added Table 1, which summarizes the characteristics of each mouse.
- Please include a summary table displaying major immune cell counts and cytokine
expression variations across all models (with and without IL-36α) would assist develop
a large dataset.
Response: Table 1 incorporates each characteristic.
- A longitudinal time-point analysis after IL-36α injection might help determine if
cytokine profiles alter dynamically over the period.
Response: We appreciate the reviewer’s insightful comment regarding the potential value of a longitudinal analysis. In our preliminary pilot experiment, we performed a single-sample analysis at a mid-point time after IL-36α injection to explore possible dynamic changes in cytokine profiles. However, we did not observe marked alterations in the lymphocyte cytokine expression profile at that time point. Based on this result, we proceeded with the current study design focusing on the endpoint analysis. We agree that a comprehensive time-course analysis could provide further insights and plan to address this in future investigations.
- While the mouse results are strong, comparing key cytokine expression patterns or
immune profiles to human AD patient biopsy or serum data (even from published
literature) would increase translational relevance. I suggest looking into this and
providing the necessary information in the MS.
Response: Thank you for pointing this out. We have added the references and changed the text.
Reviewer 2 Report
Comments and Suggestions for Authors
The manuscript provides valuable insights into the role of IL-36α in AD. With the inclusion of these additional analyses and references, the manuscript is well positioned for acceptance:
- Endogenous Levels of IL-36α:
It would be valuable to assess the endogenous levels of IL-36α in the skin to better understand its natural role in AD before the introduction of exogenous IL-36α. This additional data could enhance the understanding of IL-36α's involvement in early inflammatory responses and help contextualize its role in disease onset. - Levels of Skin Barrier Markers and Their Correlation with IL-36α:
Exploring gene expression related to skin barrier function and keratinocyte differentiation could strengthen the study’s conclusions. Moreover, it would provide deeper insights into how IL-36α influences skin barrier dysfunction, a hallmark of AD. - Suggested Literature:
- Zysk W., et al. (2023) suggest that IL-36α may serve as a biomarker to distinguish lesional from non-lesional skin in AD.
- Balato A., et al. (2022) offers a comprehensive overview of biomarkers in AD and could help compare the findings of this study with other biomarkers influencing both AD and psoriasis. Including these references would enrich the manuscript's discussion by highlighting common aspects of both diseases and provide further support for the manuscript’s discussion of IL-36α as a potential biomarker in AD.
References:
- Zysk W., et al. Altered Gene Expression of IL-35 and IL-36α in the Skin of Patients with Atopic Dermatitis. Int J Mol Sci, 2023.
- Balato A., et al. The Impact of Psoriasis and Atopic Dermatitis on Quality of Life: A Literature Research on Biomarkers. Life (Basel), 2022.
Author Response
The manuscript provides valuable insights into the role of IL-36α in AD. With the inclusion of these additional analyses and references, the manuscript is well positioned for acceptance:
- Endogenous Levels of IL-36α: It would be valuable to assess the endogenous levels of IL-36α in the skin to better understand its natural role in AD before the introduction of exogenous IL-36α. This additional data could enhance the understanding of IL-36α's involvement in early inflammatory responses and help contextualize its role in disease onset.
Response: Thank you for your suggestion. We have examined the intrinsic level of IL-36α, β, and γ in the skin and the change after the administration of IL-36a. The results are shown in Figure 5A.
- Levels of Skin Barrier Markers and Their Correlation with IL-36α:
Exploring gene expression related to skin barrier function and keratinocyte differentiation could strengthen the study’s conclusions. Moreover, it would provide deeper insights into how IL-36α influences skin barrier dysfunction, a hallmark of AD.
Response: Thank you very much for your excellent comments. We have examined the changes in loricrin, involucrin, and filaggrin levels before and after administering IL-36a. The results are shown in Figure 5B.
- Suggested Literature:
Zysk W., et al. (2023) suggest that IL-36α may serve as a biomarker to distinguish lesional from non-lesional skin in AD.
Balato A., et al. (2022) offers a comprehensive overview of biomarkers in AD and could help compare the findings of this study with other biomarkers influencing both AD and psoriasis. Including these references would enrich the manuscript's discussion by highlighting common aspects of both diseases and provide further support for the manuscript’s discussion of IL-36α as a potential biomarker in AD.
Response: We have cited the references you mentioned in the text.
Round 2
Reviewer 1 Report
Comments and Suggestions for Authors
Dear Authors,
The revised manuscript seems to be improved as per suggested. The manuscript can be considered for publication in its current form.
Author Response
Response: We sincerely thank Reviewer 1 for the positive feedback and are grateful that the manuscript is considered acceptable for publication.Reviewer 2 Report
Comments and Suggestions for Authors
The authors have improved the manuscript as suggested. However, in some plots, the median and standard deviation are not shown. Please adjust them consistently, as done in the other plots.
Author Response
Response: Thank you for your comment. We have adjusted all plots to display the median and standard deviation (SD) consistently. In some cases, the SD extends beyond the axis limits, and we have added a note in the figure legend to clarify this.